# Phytochemical, Technological, and Pharmacological Study on the Galenic Dry Extracts Prepared from German Chamomile (*Matricaria chamomilla* L.) Flowers

**DOI:** 10.3390/plants13030350

**Published:** 2024-01-24

**Authors:** Janne Sepp, Oleh Koshovyi, Valdas Jakstas, Vaidotas Žvikas, Iryna Botsula, Igor Kireyev, Karina Tsemenko, Oleksandr Kukhtenko, Karin Kogermann, Jyrki Heinämäki, Ain Raal

**Affiliations:** 1Institute of Pharmacy, Faculty of Medicine, University of Tartu, Nooruse 1, 50411 Tartu, Estonia; janne.sepp@ut.ee (J.S.); oleh.koshovyi@ut.ee (O.K.); karin.kogermann@ut.ee (K.K.); jyrki.heinamaki@ut.ee (J.H.); 2Pharmacognosy Department, National University of Pharmacy, 53 Pushkinska Str., 61002 Kharkiv, Ukraine; 3Institute of Pharmaceutical Technologies, Lithuanian University of Health Sciences, 44307 Kaunas, Lithuania; valdas.jakstas@lsmu.lt (V.J.); vaidotas.zvikas@lsmu.lt (V.Ž.); 4Pharmacology and Pharmacotherapy Department, National University of Pharmacy, 53 Pushkinska Str., 61002 Kharkiv, Ukraine; botsula.iv@gmail.com (I.B.); ivkireev@ukr.net (I.K.); k-cemenko@ukr.net (K.T.); 5Pharmaceutical Technology of Drugs Department, National University of Pharmacy, 53 Pushkinska Str., 61002 Kharkiv, Ukraine; kukhtenk@gmail.com

**Keywords:** German chamomile, flower, galenic extract, extraction, complex processing, analgesic activity, soporific effect

## Abstract

Galenic preparations of German chamomile are used to treat mild skin diseases, inflammation, and spasms, and they have also been reported to have anxiolytic and sedative effects. The medicinal use of chamomile is well known in ethnomedicine. After obtaining its galenic preparations, there is lots of waste left, so it is expedient to develop waste-free technologies. The aims of this study were to gain knowledge of the ethnomedical status of chamomile in the past and present, develop methods for preparing essential oils and dry extracts from German chamomile flowers using complex processing, reveal the phytochemical composition of such extracts, and verify the analgesic and soporific activity of the extracts. Two methods for the complex processing of German chamomile flowers were developed, which allowed us to obtain the essential oil and dry extracts of the tincture and aqueous extracts as byproducts. A total of 22 phenolic compounds (7 hydroxycinnamic acids, 13 flavonoids, and 2 phenolic acids) were found in the dry extracts by using UPLC-MS/MS. In total, nine main terpenoids were identified in the chamomile oil, which is of the bisabolol chemotype. During the production of chamomile tincture, a raw material–extractant ratio of 1:14–1:16 and triple extraction are recommended for its highest yield. In in vivo studies with mice and rats, the extracts showed analgesic activity and improvements in sleep. The highest sedative and analgesic effects in rodents were found with the dry extract prepared by using a 70% aqueous ethanol solution for extraction at a dose of 50 mg/kg. The developed methods for the complex processing of German chamomile flowers are advisable for implementation into the pharmaceutical industry to reduce the volume of waste during the production of its essential oil and tincture, and to obtain new products.

## 1. Introduction

Since ancient times, German chamomile has been used in both folk and official medicine. Chamomile (*Matricaria chamomilla* L.) belongs to the family *Asteraceae* and is an essential-oil-containing medicinal herb that is widely known and used in Europe, Asia, and the Americas [1,2,3]. Chamomile is usually consumed as a tea or tincture. Its essential oil and tincture are the components of several traditional and homeopathic medicinal drugs [1,4]. To date, the galenic preparations of German chamomile have been used to treat mild skin diseases, inflammation, and spasms, and such preparations have also been reported to have anxiolytic and sedative effects [1], but there is less scientific proof of their analgesic and soporific activity to date. They are also useful in treating many other diseases and disorders, such as flatulence, colic, ulcers, wounds, hysteria, depression, etc. [1]. In 2000, chamomile was assigned as an over-the-counter (OTC) dietary supplement by the US FDA. In addition, German chamomile and its essential oil, extracts, and distillates are generally regarded as safe for use in food products [1]. The national pharmacopoeia of a total of 26 countries recognizes chamomile as a drug [1,4]. Since chamomile is widely used for the treatment of various diseases, its pharmaceutical and medicinal value cannot be ignored. Therefore, it is worth developing novel pharmaceutical formulations and complex technologies based on galenic preparations of chamomile to further improve the rational medicinal use and efficacy of this medicinal herb.

Galenic preparations and dry raw materials of German chamomile are widely applied as medicines. One of the most common medicinal preparations of this herb is chamomile tinctures. The method for the preparation of chamomile tinctures is well known [5]; however, it is not optimized. The main limitation of the tincture is poor chemical stability and, consequently, the changes in the pharmacodynamics of its active components in medicinal uses. In addition, ethanol is used as an extracting solvent in the preparation of chamomile tinctures, thus excluding many patient groups who cannot use this medicinal product (e.g., children, pregnant women, nursing mothers, persons whose activities require increased attention, etc.). Ethanol can also directly affect the central nervous system (CNS) and modulate the effects of the active ingredients of the medicinal herb in question. Therefore, chamomile decoctions and teas are recommended for the abovementioned patients. Decoctions and teas, however, also have limitations in terms of medicinal uses, since they are non-standardized preparations. Furthermore, the preparation of both decoctions and teas takes a long time, and the chemical composition of such liquid preparations can change during storage (in a limited storage time). Therefore, there is a true demand for new standardized galenic preparations (i.e., dry extracts) based on chamomile tinctures and decoctions to advance the medicinal use of chamomile.

Today, global natural resources are becoming limited. Therefore, the interest in the development and preparation of new plant-based materials and products for medicinal uses by using complex processing and waste-free technologies is steadily growing. This approach makes it possible to extend the selection of medicines, to use natural resources rationally, to increase the profitability of pharmaceutical companies, and to reduce the negative impact of pharmaceutical production on the environment [6,7,8]. For preparing a chamomile tincture, one-time (single) extraction is used [5,9], and there are still significant amounts of biologically active substances (BASs) in the waste. Therefore, it is important to find out and use the optimal process parameters (such as the extraction frequency and the ratio of the raw material to the extractant) in the extraction of BASs from chamomile flowers, and to develop its complex processing technologies. This is also valid for the isolation and preparation of chamomile essential oil by means of hydrodistillation. The distillation liquid is usually also waste, despite the fact that it contains significant amounts of BASs. Thus, the development of standardized dry extracts of German chamomile flowers using complex raw material processing technologies is still one of the key topic areas in the pharmaceutical formulation research of this medicinal herb.

The aim of the present study was threefold: (1) to develop methods for preparing essential oils and dry extracts from German chamomile flowers using complex raw material processing technologies, (2) to disclose the phytochemical composition of such dry extracts, and (3) to verify the analgesic and soporific activity of these extracts in vivo with mice and rats. Moreover, this study provides knowledge of the ethnomedical status of chamomile in the past and present.

## 2. Results

### 2.1. Ethnomedical Study

In the data of the Estonian Literary Museum’s folklore archive, there were a total of 150 index cards in the catalogue of chamomiles. Unfortunately, it was not possible to distinguish the two species (German chamomile and pineapple weed) of the genus *Matricaria* (chamomile), which have been and still are common plants in Estonia. Therefore, both species of the genus are discussed here together. The materials analyzed date back to 1891, thus covering a period of almost 100 years (the latest entry is from 1989). All Estonian counties (n = 15) were represented in this study according to the current administrative division. Almost one-third of the records (index cards) did not specify the medical use of chamomiles, and the same proportion referred to the treatment of respiratory tract diseases. According to the index cards, chamomiles were used for relieving ocular diseases (15.3%), inflammation (6.0%), and trauma (5.3%) (Table 1). Chamomiles were also used for treating pain and infectious diseases, for sedative effects, and for relieving spasms. There were single references to the other diseases, which were listed under the category “Other”. It is worth mentioning that two out of three descriptions of the use of chamomiles for sedative purposes were related to the treatment of children:

“*Chamomile tea was given to children against crying*”.(ERA II 201, 105 (67), written down by Lepp, K., 1938, Saare County, Karja, Leisi municipality).

“*If the children scream a lot, chamomile tea is given to the children to drink*”.(E57311 (29), written down by Eisen, M. J., year unknown, Rapla County, Vigala).

“*Chamomile was collected for tea. A sick person was given chamomile tea for sedation*”.(KKI, KS, Jõulmaa, H., 1977, East-Viru County, Iisaku, Uhe).

The most common route of administration of chamomiles (40.7%) was an external route (internal route: 36.0%). The route of administration was not specified in 12.6% of the references; alternatively, both routes of administration were mentioned (10.7%).

### 2.2. Phytochemical Composition of Dry Extracts and Essential Oil

The dry extracts of German chamomile were light powders with a light-to-dark brown color and a specific smell. The loss-on-drying (LOD) values for the extracts ranged from 4.3% to 5.0%. 

The main phenolic compounds of the dry extracts were identified and quantified by UPLC-MS/MS (Table 2). For the assay of phenolic compounds, hydrocinnamic acids, and flavonoids, the established pharmacopeia spectrophotometric methods were used. A total of 22 phenolic compounds (7 hydroxycinnamic acids, 13 flavonoids, and 2 phenolic acids) were found in the dry extracts of German chamomile (Table 2).

Table 3 lists the main terpenoids and their contents in the German chamomile essential oil obtained by a hydrodistillation method. A total of eight terpenoids were identified and quantified, representing 98.72% of the composition of the essential oil.

### 2.3. Optimization of Dry Extract G2’s Preparation

For optimizing and rationalizing an extraction process, the identification and setting of the levels of critical material and process parameters are of primary importance. Such parameters could include the nature and concentration of an extractant, the extraction conditions (i.e., temperature, speed and time, and kind of extraction), the ratio of an extractant to the raw material, etc. These parameters affect not only the pharmaceutical quality of the product but also the manufacturing costs and, subsequently, the cost of the final medicinal product. For example, the ineffective use of extractants can significantly increase the formulation-related challenges in pharmaceutical development and reduce the therapeutic effect of the medicinal product.

To determine an effective ratio of extractant to German chamomile flowers in preparing a chamomile tincture, the yield of extractive substances and BASs (i.e., phenolic compounds, hydroxycinnamic acids, and flavonoids) was studied based on DIR (the ratio of extractant to raw materials) and the multiplicity of extraction. For this study, the maceration was performed by using 70% aqueous ethanol solution and conducting a total of six sequential stages of extraction. The volume of ethanol used in the extraction was varied to rationalize the preparation of a chamomile tincture. The study was conducted at ambient room temperature under normal pressure conditions using a laboratory percolator. In the liquid extracts, the contents of extractive substances and BASs (i.e., phenolic compounds, hydroxycinnamic acids, and flavonoids) were determined using pharmacopeia methods (Table 3). A total of 500.0 g of German chamomile flowers was used as the plant material. The coefficient for the absorption of the extractant was 2.04 (chamomile flowers in a 70% aqueous ethanol solution). The results of the extractive substance contents in the dry residue are summarized in Table 4.

Based on the phytochemical results (as shown in Table 4), the yield of extractive substances and BASs (i.e., phenolic compounds, hydroxycinnamic acids, and flavonoids) was used as a major criterion for optimizing the chamomile extraction process. For optimizing the BAS extraction rate, the mass yield coefficient of each stage (m_i BAS_/V_i extractant_) was calculated for each of the indicators. The dependence of these factors on the extraction rate was derived to determine a rational extraction rate (Figure 1) [12,13].

The effects of the DIR and the multiplicity of extraction on the contents of extractive substances (i.e., yield) were determined. Figure 1 shows the effects of the DIR ratio on the yield of extractive substances and BASs (i.e., phenolic compounds, hydroxycinnamic acids, and flavonoids) in the extraction of German chamomile flowers with a 70% aqueous ethanolic solution. Polynomial equations of the dependence between the yield of the BASs and the ratio of the extractant to the raw material were generated and used for the optimization and rationalization.

### 2.4. Pharmacological Study on Analgesic and Soporific Activity

#### 2.4.1. Analgesic Activity

Galenic preparations of chamomile are widely used for the treatment of inflammatory skin diseases and dental disorders due to their anti-inflammatory and analgesic effects. In the present study, the analgesic activity of chamomile dry extracts was verified with a hot-plate test.

Table 5 shows the results of a hot-plate test with mice, confirming the analgesic effect of the German chamomile extracts studied. The administration of dry extracts G1 and G3 slightly increased the reaction time of mice to the thermal stimulus. A greater analgesic effect was observed with the mice administered with dry extract G2 (at all three doses studied) compared to a group of reference mice, and the mice received acetaminophen.

#### 2.4.2. Soporific Activity

Chamomile tea has been used for centuries in sedative applications and for improving sleep. In our study, the soporific activity of chamomile dry extracts was studied with rats by determining the duration of sleep (i.e., the time for which the rats were in a lateral position) after the administration of the dry extracts and sodium thiopental. The results of the soporific activity are summarized in Table 6. The administration of German chamomile extracts 20 min before sodium thiopental intake induced a prolonged sleep effect in rats. With the animal group given the extract G2 (at a dose of 50 mg/kg) prior to the administration of sodium thiopental, the sleeping time was prolonged by 117.3% compared with the animal group given sodium thiopental (40 mg/kg) only. This suggests the sedative effect of the present dry extract of chamomile.

## 3. Discussion

Our previous works have shown ethnomedicinal traditions to be a valuable and inspired source of ideas for pharmaceutical studies [14,15,16]. In total, three folkloristic descriptions were found in the ethnomedicine database of Estonia on the use of chamomile tea as a sedative aid [17,18]. Interestingly, two of these descriptions are related to children, stating the following indications: “to children against crying” and “if the children scream a lot”. The reason for crying and/or screaming may be meteorism after feeding the children. On the other hand, the Historical Estonian Folk Medicine Botanical Database (“Herba”) [19] shows that the main sedative plants used in Estonia are chamomile, valerian, and lime flowers. The “Herba” database also suggests “If there is a lack of sleep, drink chamomile tea.” In general, the use of chamomile for the treatment of diseases related to the CNS has not been very common in Estonian folk medicine. However, there are some descriptions related to the use of chamomile for curing CNS diseases, and this information is confirmed by scientific studies. Numerous clinical trials have shown the sedative and hypnotic effects of chamomile, thus supporting the treatment of anxiety, depression, and insomnia [20].

In the present study, two methods for the complex processing of German chamomile flowers were explored. The first one allowed us to obtain the dry extract based on the chamomile tincture and the dry aqueous extract from the tincture waste, while the second one supposed the production of the chamomile essential oil and the dry extract based on the distillation liquid. Previously, only the tincture or the essential oil was produced from the raw material, and then all of the waste would be thrown away [9,17]. The dry extracts were hygroscopic powders of a light brown color with a specific smell. The proposed methods allowed us to obtain new dry extracts with analgesic and soporific activity from the wastes of the chamomile tincture and essential oil production.

In our study, a total of 22 phenolic compounds were found in the German chamomile extracts, and of these compounds, hydroxycinnamic acids were the predominant ones. The major hydroxycinnamic acids in the present dry extracts were 4,5-dicaffeoylquinic acid, 3,5-dicaffeoylquinic acid, 3,4-dicaffeoylquinic acid, chlorogenic acid, and neochlorogenic acid. In our chamomile dry extracts, however, the predominant substances were not the same as described in the literature [21,22]. Mulinacci et al. [22] reported that the extracts of chamomile flowers contained 39% cinnamic acid derivatives, such as ferulic acid and caffeic acid.

In the dry extracts of German chamomile flowers, luteolin and quercetin derivatives were found as the predominant flavonoids, and apigenin, kaempferol, and isorhamnetin derivatives were present in smaller amounts. The most dominant flavonoids were luteolin-7-O-glucoside and isoquercitrin. Recently, Catani et al. [21] reported that the main flavonoids present in the German chamomile raw material were apigenin, quercetin, patuletin, and luteolin, at concentrations of 16.8%, 9.9%, 6.5%, and 1.9%, respectively. The results reported by Catani et al. [21] are not in line with our findings in the current study.

It should be noted that the significant amount of 3,4-dihydroxyphenylacetic acid in the extract could be related to analgesic activity [23].

In chamomile essential oil, a total of eight main compounds were quantified, representing more than 98% of the total oil. Table 3 shows the RI values of the principal compounds in the two columns representing different polarity and concentration ranges (>1%). (E)-ß-farnesene (25%), α-bisabolol oxides A and B (both 22%), and α-bisabolone oxide A (10%) were the main compounds in the essential oil. The content of chamazulene was less (8%). Based on its chemical contents, the present chamomile essential oil belongs to the chemotype rich in bisabolol oxides described in the European Pharmacopoeia [24]. Previously, we studied the essential oil of pineapple weed (*Matricaria discoidea* DC.), which is similar to German chamomile oil but is low in bisabolol oxides [25].

According to the literature, the main components of German chamomile are terpenoids, such as α-bisabolol and its oxides, azulenes, chamazulene (1–15%), and apigenin [1,26,27,28]. The phenolic composition of our extracts, however, differed from the findings reported in the scientific literature.

We found that in preparing the dry extracts of chamomile the most effective ratio of extractant to raw materials was in the range of DIR 1:16–1:18 (Figure 1). With this ratio, the contents of extractive substances reached a “plateau” and were not significantly increased with an increase in the amount of extractant used. When extracting hydroxycinnamic acids and flavonoids, it is advisable to use a ratio of extractant to raw material in the range of DIR 1:12–1:14. Increasing the amount of the extractant does not significantly increase the output of these BASs. At the same time, it is more appropriate to extract phenolic compounds in the range of DIR 1:14–1:16. In summary, for the highest yield of phenolic compounds in the extract, the ratio of extractant to raw material should be 1:14–1:16, and a triple extraction is recommended.

The present study demonstrates that the administration of German chamomile extracts via an intragastric route enhances analgesic effects in mice in a hot-plate test. The administration of the extracts G1 and G3 resulted in a slight prolongation of the time of discomfort. The administration of extract G1 at doses of 50 mg/kg and 100 mg/kg extended the time period that the mice spent on the test plate (one hour after administration and before the occurrence of the discomfort reaction) by 38% and 43% (*p* < 0.05), respectively. Interestingly, the administration of extract G3 increased the corresponding time period only by 17% and 25%, respectively. The comparison was made with the corresponding time period observed for the control group of intact animals.

The administration of extract G2 to mice at all three doses studied, and at different time points, resulted in a significant (*p* < 0.05) analgesic effect in the rodents in comparison to the control group of intact animals and the group of animals that received acetaminophen. The maximum analgesia was found at 60 min after administration. The time of discomfort occurrence in the animal group that received the extract at doses of 50 mg/kg and 100 mg/kg was 67% and 74% higher compared to that observed in the control group, respectively. Moreover, the analgesic activity of these extracts at the doses studied and one hour after the administration was increased by 14% and 28%, respectively. The comparison was made with the analgesic activity found in the animal group treated with acetaminophen.

The present results obtained via the hot-plate test in mice show the analgesic activity of the German chamomile extracts studied. In the recent study reported by Chaves et al. [29], the analgesic activity of a crude chamomile fraction was investigated with a formalin test. The authors reported that reduced nociception (by 96%) was observed upon using a 30 mg/kg dose compared to the control (10 mL/kg of saline solution), thus demonstrating analgesic properties. These findings are consistent with our results. To the best of our knowledge, no other studies have been published on the analgesic activity of German chamomile preparations in the state-of-the-art literature.

The vast majority of the studies published to date have reported the anti-inflammatory activity of chamomile products [1], while little is known about their analgesic activity accompanied by anti-inflammatory activity. Lee et al. [30] studied the applicability and efficiency of German chamomile fixed oil with an atopic dermatitis animal model (mice) [30]. After the administration period of 4 weeks, there was a significant reduction in serum IgE and IgG1 levels in the mice. Bhaskaran and co-workers [31] investigated dried chamomile flower extracts and their mechanisms of action in inflammatory disorders. The role of luteolin (flavonoid) in generating the anti-inflammatory effects of chamomile has also been studied and discussed [32]. Flemming et al. [33] investigated the anti-inflammatory activity of matricin from chamomile flowers in vivo with mice using carrageenan-induced inflammation and air-pouch models. The results showed a significant dose-dependent increase in anti-inflammatory responses in mice. The potential effects of matricin and chamazulene on inflammation were studied [33]. Some earlier studies also reported the anti-inflammatory activity of essential oil components, such as α-bisabolol, bisabolonoxid [34], and polyketides [35]. However, there is still very limited information about the analgesic activity of chamomile and its preparations.

Chamomile tea has been used for centuries for inducing calmness and for the treatment of sleep disorders [1]. It has been reported that the sedative effect is mainly due to the action of the flavonoid apigenin, found in chamomile [1,36]. Apigenin acts by binding to benzodiazepine receptors present in the brain.

Sodium thiopental (a barbiturate) induces sleep efficiently in both humans and rodents. A sodium-thiopental-induced sleeping test is widely used for the study of the sedative/soporific activity of new active pharmaceutical ingredients [37]. We found that the German chamomile extracts (G1–G3) had a sedative effect of 30.5–117.3% on rats compared to the control group. With the rat group that received extract G1, the average duration of sleep was 140.3 ± 6.5 min at the dose of 25 mg/kg, 201.8 ± 4.7 min at the dose of 50 mg/kg, and 148.8 ± 3.9 min at the dose of 100 mg/kg. These values for sleeping time period are significantly higher (by 33.9%, 92.5%, and 42.0%, respectively) compared to the corresponding time periods observed with the control group (*p* < 0.05). The duration of sleep was prolonged in all groups that received extract G3, by 69.6%, 30.5%, and 58.7%, respectively.

The administration of extract G2 at a dose of 50 mg/kg resulted in the highest sedative effect in rats; consequently, the duration of sleep increased by 117.3% compared to that observed in the control group. The duration of sleep was also longer than the average sleeping period in a reference group that received valerian syrup. Reducing the dose of extract G2 to 25 mg/kg led to a slight decrease in sedative activity (77.7%) in rats compared to the control group (*p* < 0.05). Increasing the dose of extract G2 from 50 mg/kg to 100 mg/kg did not have any influence on the sedative activity in rats in comparison to a rodent group treated with extract G2 at a dose of 50 mg/kg. This suggests that the administration of extract G2 at a dose of 50 mg/kg provides a sedative action in rats, with a pronounced pharmacodynamic effect.

We found that the German chamomile extracts exhibit a synergistic soporific effect with sodium thiopental. A dose-dependent decrease in locomotion was observed in the mice, and the maximum effect was achieved at a dose of 30 mg/kg of the chamomile crude fraction. Mice usually demonstrate anxiolytic activity by burying noxious things. Several earlier studies have reported the anti-anxiety effects of chamomile products [29]. Moreover, chamomile and its preparations could affect fluctuations in cortisol levels associated with CNS disorders [38]. Carpenter et al. [39] reported that elevated levels of adrenocorticotropic hormone (ACTH) are associated with stress and anxiety. Yamamoto et al. [40] found that chamomile extracts possess neurokinin-1 receptor antagonist activity [39]. Furthermore, the inhalation of chamomile oil vapors was shown to reduce the ACTH levels caused by the stress induced by ovariectomy in rats [41]. Recently, Amsterdam et al. [42] reported that flavonoid components in chamomile modulate central neurotransmitter activity (i.e., reductions in serotonin, dopamine, and monoamine oxidase activity) while elevating catecholamine production and noradrenalin activity. In addition, chamomile possesses ingredients that play an important role in CNS diseases, such as epilepsy and Alzheimer’s disease. In the study carried out by Hashemi and co-workers, convulsions were induced by the administration of kainic acid [43]. To date, numerous studies have been published on the effects of chamomile products on the CNS [1]. However, little is known about chamomile’s effect on healthy users’ sleep. This shows the novelty of our study.

## 4. Materials and Methods

### 4.1. Ethnomedical Study

The Estonian Literary Museum is a research and development institution managing important archives of cultural history and folklore. In the ethnomedicine of Estonia, the plants from the genus *Matricaria* (chamomile) have been used to relieve various health problems. The ethnomedicinal catalogue entitled “Chamomile” belongs to the card index “Ethnobotany”. In our study, we used these established sources of information, and a categorized electronic dataset was constructed to enable further analysis.

### 4.2. Plant Material

German chamomile flowers (1.0 kg) were collected from the herb company MK Loodusravi OÜ’s field of medicinal plants located in the Päpe farm, Venevere village, Põhja-Sakala municipality, Viljandi County, Estonia (58.598228 N, 25.704950 E). The flowers were dried at 30–35 °C and stored in airtight plastic for further studies. The identity of the raw material was established by Prof. Ain Raal, Institute of Pharmacy, University of Tartu, Tartu, Estonia [25,44,45]. The raw material was standardized according to the European Pharmacopeia’s requirements [24]. The loss on drying of the flowers was 6.8% [24].

### 4.3. Preparation of Extracts

A total of 100.0 g of the dried German chamomile flowers [24] was added to 1250.0 mL of water and distilled in the essential oil extraction device (Albrigi Luigi SRL, Stallavena, Italy) for 3 h to obtain the essential oil. The content of the essential oil was 5 mL/kg in the dry raw material. After cooling, the aqueous distillated extract was separated from the raw material by paper filtration. The volume of the extract was 705 mL. The dry residue of the extract was 4.1 ± 0.4%. The distilled extract was evaporated to a dry extract (extract G1) by lyophilic drying in a SCANVAC COOLSAFE 55-4 Pro (LaboGene ApS, Lillerød, Denmark) apparatus. The yield of dry extract G1 was 28.9%.

A total of 500.0 g of the dried German chamomile flowers [24] was macerated with 3000 mL of 70% aqueous ethanol solution in an extractor at ambient room temperature overnight. The process was repeated five more times with 1000.0 mL of the same solvent to determine the recommended and rational multiplicity of extraction. The first three liquid extracts were combined, kept for sedimentation for two days, and finally filtrated. The liquid extract was then evaporated with a rotary vacuum evaporator to a thick extract, which was dried by lyophilic drying in a SCANVAC COOLSAFE 55-4 Pro (LaboGene ApS, Denmark) apparatus. The yield of dry extract G2 was 31.9%.

The waste of the raw material (after a three-step extraction with 70% aqueous ethanol solution) was mixed with 1000.0 mL of water and boiled for 30 min. After cooling, the aqueous extract was separated from the raw material by paper filtration and evaporated to a dry extract (extract G3) by lyophilic drying in a SCANVAC COOLSAFE 55-4 Pro (LaboGene ApS, Denmark) apparatus. The yield of dry extract G3 was 7.5%.

### 4.4. Phytochemical Analysis

#### 4.4.1. Assay of Main Phytochemicals

The quantification of hydroxycinnamic acids, flavonoids, and total phenols in the chamomile extracts was performed with a Shimadzu UV-1800 (Shimadzu Corporation, Tokyo, Japan) spectrophotometer. Hydroxycinnamic acids were determined in terms of chlorogenic acid at 525 nm after reaction with hydrochloric acid, sodium nitrite, and sodium molybdate [24]. Flavonoids were assayed in terms of rutin at 417 nm after the formation of the complex with aluminum chloride [7,24]. The contents of total phenolic compounds were determined in terms of gallic acid at 270 nm [45]. For statistical validity, the experiments were performed in triplicate.

#### 4.4.2. Gas Chromatographic Analysis of Essential Oil

The main compounds (>1%) of chamomile essential oil were analyzed using Agilent’s GC 7890a chromatograph (Santa Clara, CA, USA) with Agilent Open Lab CDS Chem Station software (Rev. C.01.07[27]) and FID. The analysis was conducted on two fused silica capillary columns with stationary phases: DB-5 and HP-Innowax (both 30 m × 0.25 mm, Agilent, CA, Santa Clara, USA). The carrier gas used was hydrogen, with a split ratio of 1:150 and a flow rate of 30 mL/min. The temperature program ranged from 50 to 250 °C at 2.92 °C/min, while the injector temperature was 250 °C.

The Agilent Open Lab CDS Chem Station software (Rev. C.01.07[27]) was used to identify the oil’s principal components and to compare their retention indices. The component contents (%) of the essential oil were determined by analyzing the mean retention time and peak area of four parallel chromatograms. We identified the components by comparing their DB-5 column retention indices to databases and literature data [9,10,11,36].

#### 4.4.3. Identification of Phenolic Compounds by UPLC-MS/MS

Determination of the phenolic compounds in German chamomile flowers was carried out with a UPLC-MS/MS system. Chromatographic separation was performed using an Acquity H-class UPLC system (Waters, Milford, MA, USA) equipped with a YMC Triart C18 (100 × 2.0 mm 1.9 µm) column. The constant temperature of the column was 40 °C during the analysis. The flow rate of the mobile phase was 0.5 mL/min. An aqueous solution of formic acid (0.1%) was used as solvent A, and MS-grade acetonitrile was used as solvent B. The following linear gradient was applied: solvent A from 0 to 1 min, isocratic conditions at 95%; 1 to 5 min, linear decrease to 70%; 5 to 7 min, to 50%; 7.5 to 8 min, wash column with 100% solvent B; 8.1 to 10 min, equilibrate column with initial conditions. Mass spectrometric analysis was carried out with a triple-quadrupole tandem mass spectrometer (Xevo, Waters, USA). Negative electrospray ionization (ESI) was performed to acquire MS/MS data. The settings for the MS/MS analysis were as follows: the voltage of the capillary tip was set to -2 kV, nitrogen gas heated to 400 °C was flowing at 700 L/h, the curtain gas flow was set to 20 L/h, and the ion source temperature was set to 150 °C. Identification of phenolic compounds in German chamomile flowers was established by comparing their retention times and MS/MS spectral data with those of commercial reference substances. The quantitative determination of phenolic compounds was performed with a standard dilution method and linear regression fit models for phenolic compounds [46].

### 4.5. Pharmacological Study

All pharmacological studies (i.e., for analgesic and soporific activity) were carried out in compliance with the rules of the “European Convention for the Protection of Vertebrate Animals Used for Experimental and Other Scientific Purposes” (Strasbourg, 1986), Directive 2010/63/EU of the European Parliament and of the Council of the European Union (2010) on the protection of animals used for scientific purposes, the Order of the Ministry of Health of Ukraine No. 944 “On Approval of the Procedure for Preclinical Study of Medicinal Products and Examination of Materials for Preclinical Study of Medicinal Products” (2009), and the Law of Ukraine No. 3447-IV “On the protection of animals from cruel treatment” (2006) [47,48,49,50,51]. The research was approved by the Bioethics Commission of the National University of Pharmacy (protocol №4 from 3 October 2023).

For the pharmacological study, the rodents (mice and rats) were kept on a standard diet and under standard conditions in the vivarium of the National University of Pharmacy (Kharkiv, Ukraine).

#### 4.5.1. Analgesic Activity

The analgesic activity of the chamomile extracts (G1, G2, G3) and the reference drug acetaminophen (paracetamol-Zdorovye, 500 mg capsules, pharmaceutical company “Zdorovye”, Kharkiv, Ukraine) was studied with mice weighing 20–40 g.

The animals were kept without food for 2 h before the test. The groups of animals were formed by the method of randomization. The period of quarantine and acclimatization lasted for 14 days. The mice were divided into 11 groups (6 mice in each group): Group 1—intact animals that received a 0.9% solution of NaCl at a dose of 1 mL per 100 g of body weight; Group 2—animals that received extract G1 at a dose of 25 mg/kg; Group 3—animals that received extract G1 at a dose of 50 mg/kg; Group 4—animals that received extract G1 at a dose of 100 mg/kg; Group 5—animals that received extract G2 at a dose of 25 mg/kg; Group 6—animals that received extract G2 at a dose of 50 mg/kg; Group 7—animals that received extract G2 at a dose of 100 mg/kg; Group 8—animals that received extract G3 at a dose of 25 mg/kg; Group 9—animals that received extract G3 at a dose of 50 mg/kg; Group 10—animals that received extract G3 at a dose of 100 mg/kg; Group 11 (control group (CG))—animals that received acetaminophen at a dose of 50 mg/kg.

The chamomile extracts studied were administered intragastrically at doses of 25 mg/kg, 50 mg/kg, and 100 mg/kg in the form of an aqueous suspension 30 min before placing the animals on the equipment. The reference drug, acetaminophen, was administered at a dose of 50 mg/kg as a solution.

After the administration of the extract or reference drug, the animal was carefully placed on a hot plate (55 °C) in 30 min. The indicator of pain sensitivity was the duration of the animal’s stay (in seconds) on the hot plate before the onset of protective reflexes (e.g., licking of limbs, rebound). The mice were observed for 0.5, 1, 2, 3, and 4 h. The criterion of the analgesic effect was a significant increase in the latent period of response after the administration of the sample (extract or drug) compared to the control. To prevent thermal damage during the experiment, the time of the animals’ exposure to the hot plate did not exceed 60 s. In the “hot plate” model, the analgesic activity was calculated according to the following equation:$$AA=\frac{Te-Tc}{Tc}\times 100\%$$
where AA is the analgesic activity (%);

Te is the difference in the latent period of the corresponding response in the group of experimental animals before and after administration of a potential analgesic;

Tc is the difference in the latent period of the corresponding response in the group of control animals before and after administration of the solvent.

Statistical data analysis was performed using parametric methods of statistics with Student’s *t*-test. The level of statistical significance of differences was *p* < 0.05.

#### 4.5.2. Soporific Activity

The soporific activity of the chamomile extracts (G1, G2, G3), sodium thiopental lyophilizate (for injection solution, PLC “Kiivmedpreparat”, Kyiv, Ukraine), and “Valerian syrup AN NATUREL” syrup (LLC Beauty and Health, Kharkiv, Ukraine) was investigated with white rats weighing 190–280 g.

The animals were divided into 12 groups (6 rats in each group): Group 1—intact animals; Group 2—animals that received extract G1 at a dose of 25 mg/kg; Group 3—animals that received extract G1 at a dose of 50 mg/kg; Group 4—animals that received extract G1 at a dose of 100 mg/kg; Group 5—animals that received extract G2 at a dose of 25 mg/kg; Group 6—animals that received extract G2 at a dose of 50 mg/kg; Group 7—animals that received extract G2 at a dose of 100 mg/kg; Group 8—animals that received extract G3 at a dose of 25 mg/kg; Group 9—animals that received extract G3 at a dose of 50 mg/kg; Group 10—animals that received extract G3 at a dose of 100 mg/kg; Group 11 (control group (CG1))—animals that received the reference drug sodium thiopental at a dose of 40 mg/kg; Group 12 (control group (CG2)) (valerian)—animals that received valerian syrup at a dose of 2.14 mg/kg. The duration of sleep was determined by the time period for which the rats were in a lateral position.

### 4.6. Statistical Analysis

The mean and standard deviation (SD) were calculated according to the Monograph “Statistical Analysis of the Results of a Chemical Experiment” of the State Pharmacopoeia of Ukraine [24,52]. The average value was established on the basis of at least three measurements in the phytochemical study and on the basis of six measurements in the pharmacological study. The values of the confidence interval were calculated using Student’s criterion limit. The data are presented as the mean ± SD [24,53].

## 5. Conclusions

In the present study, two methods for preparing the essential oil and dry extracts from German chamomile flowers using complex processing were introduced: the first one allowed us to obtain the dry extracts based on the chamomile tincture and the dry aqueous extract from the tincture waste; the second one supposed the production of the chamomile essential oil and the dry extract based on the distillation liquid. A total of 22 phenolic compounds were identified and quantified in the dry extracts of German chamomile flowers. A total of nine terpenoids were identified and quantified in the essential oil of German chamomile flowers. The contents of the main phenolic compounds were successfully determined by means of spectrophotometry. The dry extracts of German chamomile flowers showed analgesic activity in a mouse model and improved sleep in a rat model. The dry extract prepared with a 70% aqueous ethanol solution showed the highest analgesic and soporific efficiency in rodents. Our study also provides evidence about the sedative use of chamomiles in Estonian folk medicine. The proposed complex raw material processing technologies for German chamomile flowers can be implemented into the pharmaceutical industry in the production of chamomile tincture and essential oil, allowing us to obtain new products from waste, use natural resources rationally, increase the profitability of pharmaceutical companies, and reduce the negative impact of pharmaceutical production on the environment.

## Figures and Tables

**Figure 1 plants-13-00350-f001:**
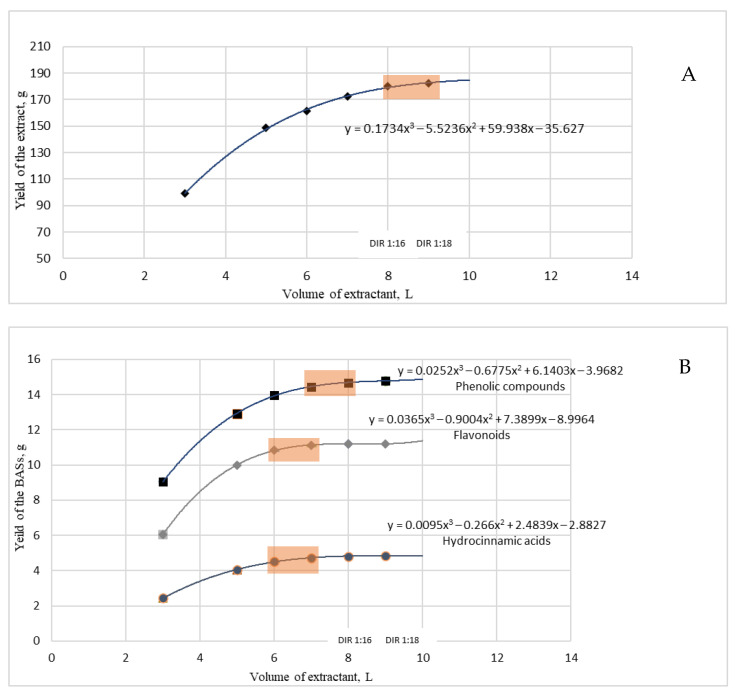
The effect of the DIR ratio on the yield of extractive substances (**A**) and BASs (**B**) in the extraction of German chamomile flowers.

**Table 1 plants-13-00350-t001:** The use of *Matricaria* plants in folk medicine in Estonia—indications.

Category	Number of Records	Percentage of Total (%)
Unspecified	45	30.0
Respiratory tract diseases	45	30.0
Ocular diseases	23	15.3
Inflammation	9	6.0
Trauma	8	5.3
Pain	5	3.3
Infectious diseases	5	3.3
Other	5	3.3
Sedative	3	2.0
Spasms	2	1.3

**Table 2 plants-13-00350-t002:** Contents of phenolic compounds in the German chamomile extracts.

Substance	Content in the Extract (X ± Δx, n = 3)		
	G1	G2	G3
UPLC-MS/MS, µg/g of dry extract			
Neochlorogenic acid	1672.30 ± 85.39	444.94 ± 20.16	441.14 ± 13.32
Luteolin	83.99 ± 12.65	310.93 ± 22.73	74.87 ± 3.871
Isoquercitrin	477.46 ± 68.82	921.16 ± 85.20	42.15 ± 14.12
Cryptochlorogenic acid	16.51 ± 2.42	80.74 ± 13.48	0
Luteolin-4-O-glucoside	16.98 ± 2.86	45.11 ± 3.67	0
Chlorogenic acid	3930.89 ± 224.37	11,742.31 ± 376.34	1280.86 ± 98.96
Quercetin	18.87 ± 1.20	172.15 ± 12.01	9.37 ± 1.03
Isorhamnetin-3-O-rutinoside	9.34 ± 0.56	15.40 ± 1.60	0
Isorhamnetin-3-glucoside	257.7 ± 27.04	410.75 ± 52.07	46.43 ± 4.38
Luteolin-3,7-diglucoside	18.11 ± 4.59	20.72 ± 1.88	0
Vanillic acid	175.96 ± 13.28	86.58 ± 5.54	71.59 ± 7.22
Caffeic acid	42.98 ± 3.82	43.04 ± 3.22	33.46 ± 2.95
3,4-Dihydroxyphenylacetic acid	376.15 ± 27.47	184.05 ± 13.38	159.71 ± 12.16
Isorhamnetin	17.99 ± 1.89	125,32 ± 12.71	15.51 ± 2.27
Apigenin	84.93 ± 4.88	578.65 ± 63.91	12.98 ± 2
Kaempherol-3-O-glucoside	34.79 ± 1.46	50.76 ± 2.10	0
Rutin	45.34 ± 5.55	126.49 ± 5.73	0
Hyperoside	224.42 ± 21.56	366.82 ± 21.21	65.15 ± 2.36
Luteolin-7-O-glucoside	616.65 ± 63.46	1061.82 ± 83.68	123.64 ± 31.82
4,5-Dicaffeoylquinic acid	3565.27 ± 266.90	4912.17 ± 416.85	541.70 ± 26.44
3,5-Dicaffeoylquinic acid	1823.72 ± 136.53	2512.69 ± 213.23	277.09 ± 13.52
3,4-Dicaffeoylquinic acid	3739.46 ± 279.94	5152.17 ± 437.22	568.17 ± 27.73
Spectrophotometry, %			
Phenolic compounds	6.19 ± 0.29	9.70 ± 0.52	2.27 ± 0.11
Hydrocinnamic acids	1.57 ± 0.09	3.47 ± 0.15	0.21 ± 0.01
Flavonoids	3.63 ± 0.11	9.92 ± 0.32	0.45 ± 0.01

Notes: G1—the dry water extract after distillation of essential oil; G2—the dry extract obtained with 70% ethanol solution; G3—the dry water extract after obtaining the tincture; *p* = 0.0996.

**Table 3 plants-13-00350-t003:** The contents of principal terpenoids in the German chamomile essential oil.

RI (DB-5)	Compound	Content in the Oil (%)	References [10,11]
1455	(E)-ß-Farnesene	24.72	1450–1456
1471	Germacrene D	1.01	1470–1478
1570	Spathulenol	2.39	1568–1570
1649	α-Bisabolol oxide B	22.27	1646–1649
1674	α-Bisabolone oxide A	10.40	1670–1675
1715	Chamazulene	7.89	1711–1715
1743	α-Bisabolol oxide A	21.78	1734–1748
1874	cis-Enyne-bicycloether	8.26	1867–1876
In total		98.72	

**Table 4 plants-13-00350-t004:** Dynamics of phenolic compounds’ extraction with 70% aqueous ethanol solution from German chamomile flowers.

Extraction Stage	Dry Residue (%)	Content (%) in the Dry Residue		
		Phenolic Compounds	Hydrocinnamic Acids	Flavonoids
1	5.00 ± 0.28	9.13 ± 0.34	2.46 ± 0.12	6.12 ± 0.19
2	2.57 ± 0.13	7.80 ± 0.10	3.24 ± 0.17	7.89 ± 0.04
3	1.4 ± 0.06	8.29 ± 0.27	3.79 ± 0.23	6.89 ± 0.26
4	1.1 ± 0.08	4.39 ± 0.24	1.91 ± 0.11	2.53 ± 0.04
5	0.8	2.85 ± 0.13	0.72 ± 0.04	0.58 ± 0.03
6	0.2	1.35 ± 0.05	0.99 ± 0.03	0.28 ± 0.01

**Table 5 plants-13-00350-t005:** Analgesic activity of the German chamomile extracts in mice (n = 6).

Agent	Group	Dose (mg/kg)	The Time of Discomfort Occurrence (Seconds)/Analgesic Activity (%) in Relation to [Control] and (Reference Drug) after Administration in				
			30 min	60 min	120 min	180 min	240 min
Intact animals	1		7.20 ± 0.29	7.10 ± 0.61	7.08 ± 0.27	7.15 ± 0.65	6.73 ± 0.94
Extract G1	2	25	7.85 ± 0.39/ [9%] (−25%) *	8.48 ± 0.39/ [19%] (−18%) *	8.60 ± 0.34/ [21%] # (−18%) *	7.97 ± 0.21/ [11%] (−16%)	7.67 ± 0.34/ [14%] (−8%)
	3	50	9.65 ± 0.45/ [34%] # (−8%)	9.83 ± 0.53/ [38%] # (−5%)	9.63 ± 0.50/ [36%] # (−9%)	8.77 ± 0.27/ [23%] (−7%)	8.23 ± 0.28/ [22%] (−1%)
	4	100	9.80 ± 0.59/ [36%] # (−6%)	10.13 ± 0.61/ [43%] # (−2%)	9.97 ± 0.59/ [41%] # (−5%)	9.32 ± 0.57/ [30%] # (−1%)	8.28 ± 0.37/ [23%] (−1%)
Extract G2	5	25	9.63 ± 0.54/ [34%] # (−8%)	9.97 ± 0.60/ [40%] # (−4%)	8.65 ± 0.48/ [22%] # (−18%)	7.98 ± 0.12/ [12%] (−16%)	8.43 ± 0.21/ [25%] (1%)
	6	50	11.43 ± 0.85/ [59%] # (9%)	11.83 ± 0.77/ [67%] # (14%)	11.72 ± 0.73/ [65%] # (11%)	11.13 ± 0.73/ [56%] # (18%)	8.67 ± 0.31/ [29%] (4%)
	7	100	12.50 ± 0.36/ [74%] # (20%) *	12.52 ± 0.31/ [76%] # (21%) *	12.47 ± 0.30/ [76%] # (18%) *	9.63 ± 0.50/ [35%] # (2%)	9.02 ± 0.39/ [34%] (8%)
Extract G3	8	25	8.32 ± 0.39/ [16%] # (−20%) *	8.88 ± 0.31/ [25%] (−14%)	8.30 ± 0.17/ [17%] # (−21%) *	8.13 ± 0.30/ [14%] (−14%)	7.72 ± 0.35/ [15%] (−8%)
	9	50	7.80 ± 0.48/ [8%] (−25%) *	8.33 ± 0.45/ [17%] (−20%)	8.33 ± 0.37/ [18%] (−21%)	7.95 ± 0.34/ [11%] (−16%)	7.70 ± 0.24/ [14%] (−8%)
	10	100	8.80 ± 0.64/ [22%] (−16%)	8.88 ± 0.65/ [25%] (−14%)	8.67 ± 0.55/ [22%] (−18%)	8.38 ± 0.54/ [17%] (−11%)	8.28 ± 0.51/ [23%] (−1%)
Acetaminophen	11	50	10.54 ± 0.73	10.38 ± 0.62	10.53 ± 0.74	9.45 ± 0.60	8.35 ± 0.36

* Statistically significant (*p* < 0.05) in comparison to the group of acetaminophen 50 mg/kg (Student’s *t*-test). # Statistically significant (*p* < 0.05) in comparison to the group of intact animals (Student’s *t*-test).

**Table 6 plants-13-00350-t006:** Impact of the extracts G1, G2, and G3 on the duration of thiopental-induced sleep, t ± Δt (n = 6).

Agent	Group	Dose (mg/kg)	Average Duration of Sleep (Min)	Soporific Effect (%)
Control group	1	40	104.83 ± 8.76	100%
Extract G1	2	25	140.33 ± 6.52 *#	133.9%
	3	50	201.83 ± 4.69 *	192.5%
	4	100	148.83 ± 3.88 *#	142.0%
Extract G2	5	25	186.33 ± 6.12 *	177.7%
	6	50	227.83 ± 7.59 *	217.3%
	7	100	190.00 ± 6.97 *	181.2%
Extract G3	8	25	177.83 ± 4.00 *#	169.6%
	9	50	136.83 ± 4.74 *#	130.5%
	10	100	166.33 ± 9.93 *#	158.7%
Valerian extract	11	2.15	204.17 ± 8.39	194.8%

* Statistically significant (*p* < 0.05) in comparison to the group that received sodium thiopental (Student’s *t*-test). # Statistically significant (*p* < 0.05) in comparison to the group that received “Valerian syrup AN NATUREL” (Student’s *t*-test).

## Data Availability

Data is contained within the article.

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
