# Peer review of "Phytochemical, Technological, and Pharmacological Study on the Galenic Dry Extracts Prepared from German Chamomile (Matricaria chamomilla L.) Flowers"

_plants, 2024, doi:10.3390/plants13030350_

Round 1
Reviewer 1 Report
Comments and Suggestions for Authors
Comments for authors:
1. Overall conclusion is missing in the Abstract section.
2. Introduction: Novelty of this study in comparison to the previously published ones should be more highlighted.
3. Application of some of the non-conventional extraction techniques such as supercritical fluid extraction instead of traditional ones would be an innovation in this work. Authors should focus on a greener approach for valorization of raw materials.
4. Statistical analysis should be included in Table 2.
5. Future perspective and practical application of the finding should be stated in the Conclusions.
6. More recent studies should be cited.
7. Please, revise and reduce 10 self-citations of the coauthor Ain Raal.
Author Response
Response to the Reviewers
Dear Reviewer 1,
Title: Phytochemical, technological and pharmacological study on the galenic dry extracts prepared from German chamomile (Matricaria chamomilla L.) flowers
The authors of the manuscript thank the reviewers for their helpful comments. We have taken all comments into account when updating the manuscript and have improved our article based on them. Changes made in the manuscript are marked in yellow. Below we present our point-by-point responses to all reviewers' comments.
Q1: Overall conclusion is missing in the Abstract section.
R1: Thank you for this remark. This information was added to the abstract. Some highlights of novelty were also added.
Q2: Introduction: The novelty of this study in comparison to the previously published ones should be highlighted more. All changes are marked in yellow.
R2: Thank you for your comment. We highlighted the novelty of our study in the introduction. All changes are marked in yellow.
The first, we highlighted that there is a true demand for developing complex raw material processing technologies of German chamomile flowers.
Second, the chamomile tincture technology is not optimized, and the using of the tincture is limited because of the content of alcohol, so it’s advisable to obtain a dry extract based on the tincture. There is also flower waste left after the tincture production, which can be used for preparing a new product.
Third, the distillation liquid is usually also a waste of chamomile essential oil production, despite the fact it contains a significant amount of biologically active substances (BAS). Thus, the development of standardized dry extracts of German chamomile flowers using complex raw material processing technologies is still one of the key topic areas in the pharmaceutical formulation research of the present medicinal herb.
Fourth, we also highlighted that there is less scientifically proven dates about the analgesic and soporific activity of chamomile products.
Q3: Application of some of the non-conventional extraction techniques such as supercritical fluid extraction instead of traditional ones would be an innovation in this work. Authors should focus on a greener approach for valorization of raw materials.
R3: Thank you for this remark. The using of non-conventional extraction techniques has sense and can increase the yield of BAS and an extract, but the idea of our research was to develop and show perspectives of using complex raw material processing technologies, that’s why we used traditional ones, which is common in the pharmaceutical industry for the production of tinctures and essential oils. We highlighted in the manuscript the idea of complex raw material processing technologies for chamomile flowers.
Q4: Statistical analysis should be included in Table 2.
R4: Thanks for the remark. We are sorry that we do not understand the Reviewer here. The statistical data (± SD) are already shown in Table 2.
Q5: Future perspective and practical application of the finding should be stated in the Conclusions.
R5: Thank you for this comment. We highlighted future perspectives and practical application of our findings in the Conclusions. Thus, the proposed complex raw material processing technologies of German chamomile flowers can be implemented into the pharmaceutical industry the chamomile tincture and essential oil production, which allowed to obtain new products from waste, use of natural resources rationally, to increase the profitability of pharmaceutical companies, and to reduce the negative impact of pharmaceutical production on the environment.
Q6: More recent studies should be cited.
R6: Thanks for the comment. We added some more recent publications and deleted some older references.
Q7: Please, revise and reduce 10 self-citations of the coauthor Ain Raal.
R7: Thank you for your comment. We have deleted references # 9, 16, 18, 19, replaced them with more recent ones.

Reviewer 2 Report
Comments and Suggestions for Authors
The article entitled: Phytochemical, technological and pharmacological study on the galenic dry extracts prepared from German chamomile (Matricaria chamomilla L.) flowers, addresses a topic of great interest to readers of Plants journal. However, it requires that the authors improve the quality and scientific contribution of their work. Below I point out some suggestions for authors:
Page 11, line 401 and 402. Optimum conditions cannot be defined with this extraction. It could only be the maximum extraction.
Discussion. The main objective of the work was: (1) to develop novel methods for preparing essential oils and dry extracts from German chamomile flowers. However, not discussed or compared with other methods such as subcritical CO2 fluid and the characteristics of the extract obtained, such as color.
“5. Conclusions. In the present study, two novel methods for preparing essential oil and dry extracts from the tincture and aqueous extracts of German chamomile flowers were introduced“. Extraction with water and solvents has already been reported, for example. Plants 2022, 11(1), 29; https://doi.org/10.3390/plants11010029
Author Response
Response to the Reviewer
Dear Reviewer 2,
Title: Phytochemical, technological and pharmacological study on the galenic dry extracts prepared from German chamomile (Matricaria chamomilla L.) flowers
The authors of the manuscript thank the reviewers for their helpful comments. We have taken all comments into account when updating the manuscript and have improved our article based on them. Changes made in the manuscript are marked in yellow. Below we present our point-by-point responses to all reviewers' comments.
Reviewer # 2: The article entitled: Phytochemical, technological and pharmacological study on the galenic dry extracts prepared from German chamomile (Matricaria chamomilla L.) flowers, addresses a topic of great interest to readers of Plants journal. However, it requires that the authors improve the quality and scientific contribution of their work. Below I point out some suggestions for authors:
Q1: Page 11, line 401 and 402. Optimum conditions cannot be defined with this extraction. It could only be the maximum extraction.
R1: Thank you for this comment. We agree with your comment and made a change “optimum” to “rational”. We tried to rationalise the process of obtaining the chamomile tincture, so we just studied parameters, which are easy to change in industrial conditions, so it just was the multiplicity of extraction and DIR (the ratio of extractant to raw materials). We used only 70 % aqueous ethanol solution, which is traditionally used in the tincture technology.
Q2: Discussion. The main objective of the work was: (1) to develop novel methods for preparing essential oils and dry extracts from German chamomile flowers. However, not discussed or compared with other methods such as subcritical CO2 fluid and the characteristics of the extract obtained, such as color.
R2: Thank you for this comment. We rephrased the aim of the work, focused on the complex processing of German chamomile flowers and added the paragraph into the discussion part: “In the present study, two ways of complex processing of German chamomile flowers were proposed. The first one allowed obtained the dry extract base on the chamomile tincture and the dry aqueous extract from the tincture waste and the second one supposed the production of the chamomile essential oil and the dry extract based on the distillation liquid. Previously only the tincture or the essential oil were produced from the raw material, then all the waste used to through away [9,18]. The dry extracts were hygroscopic powders of light brown color with a specific smell. The proposed methods allowed to get new dry extracts with analgetic and soporific activity from the wastes of the chamomile tincture and essential oil production.”
We didn’t use other extraction techniques, because the main idea of our research was to develop and show perspectives of using complex raw material processing technologies for German chamomile flowers, that’s why we used traditional ones, which is common in the pharmaceutical industry for the production of tinctures and essential oils.
Q3: “5. Conclusions. In the present study, two novel methods for preparing essential oil and dry extracts from the tincture and aqueous extracts of German chamomile flowers were introduced“. Extraction with water and solvents has already been reported, for example. Plants 2022, 11(1), 29; https://doi.org/10.3390/plants11010029
R3: Thank you for this comment, we agree with this. We’ve rewritten the Conclusions and highlighted that complex processing methods of German chamomile flowers were introduced.

Reviewer 3 Report
Comments and Suggestions for Authors
The manuscript entitled 'Phytochemical, technological and pharmacological study on the galenic dry extracts prepared from German chamomile (Matricaria chamomilla L.) flowers' contains information about the phenolic and terpenoid profiles of three medicinal plant extracts, as well as their analgesic and soporific activities. In addition, the authors performed a brief ethnobotanical study on the use of this plant in Lithuania. First of all, my opinion is that this manuscript would rather fall into the scope of some pharmaceutical journal than to the Plants. Then, there is a lack of novelty that would attract the readers of this journal. The authors claim they describe new preparation protocols, but they used standard hydrodistillation and maceration which are described and used elsewhere. Also, it is quite strange authors did not detect any chamazulene in the essential oil. That should be discussed. Unfortunately, I have to suggest the rejection of the manuscript and encouragement for its submission to some other more appropriate journal.
Author Response
Response to the Reviewers
Dear Reviewer 3,
Title: Phytochemical, technological and pharmacological study on the galenic dry extracts prepared from German chamomile (Matricaria chamomilla L.) flowers
The authors of the manuscript thank the reviewers for their helpful comments. We have taken all comments into account when updating the manuscript and have improved our article based on them. Changes made in the manuscript are marked in yellow. Below we present our point-by-point responses to all reviewers' comments.
Reviewer # 3: The manuscript entitled 'Phytochemical, technological and pharmacological study on the galenic dry extracts prepared from German chamomile (Matricaria chamomilla L.) flowers' contains information about the phenolic and terpenoid profiles of three medicinal plant extracts, as well as their analgesic and soporific activities. In addition, the authors performed a brief ethnobotanical study on the use of this plant in Lithuania. First of all, my opinion is that this manuscript would rather fall into the scope of some pharmaceutical journal than to the Plants. Then, there is a lack of novelty that would attract the readers of this journal. The authors claim they describe new preparation protocols, but they used standard hydrodistillation and maceration which are described and used elsewhere. Also, it is quite strange authors did not detect any chamazulene in the essential oil. That should be discussed. Unfortunately, I have to suggest the rejection of the manuscript and encouragement for its submission to some other more appropriate journal.
Response: Thank you for your opinion and comments. To date, we have highlighted the novelty of our research in the manuscript. We proposed complex processing technologies for German chamomile flowers and pointed to perspectives of their use for obtaining new dry extracts with analgesic and soporific activity from the wastes. The first one allowed obtaining the dry extract based on the chamomile tincture and the dry aqueous extract from the tincture waste, and the second one supposed the production of the chamomile essential oil and the dry extract based on the distillation liquid. We didn’t report that we had used new methods of extraction. Chamomile essential oils are of two chemotypes: chamazulene and bisabolene. We got and studied the second one, but the content of chamazulene is also shown in the modified Table 3, which was not mentioned in the first version due to the not-so-high concentration.

Round 2
Reviewer 1 Report
Comments and Suggestions for Authors
The authors improved the quality of manuscript. Statistical analysis should be included in Table 2 in terms of statistical significance of the results obtained between G1, G2 and G3.
Author Response
Response to the Reviewer 1
Dear Reviewer,
Title: Phytochemical, technological and pharmacological study on the galenic dry extracts prepared from German chamomile (Matricaria chamomilla L.) flowers
Comment 1: The authors improved the quality of manuscript. Statistical analysis should be included in Table 2 in terms of statistical significance of the results obtained between G1, G2 and G3.
Response 1: Thank you for this clarification. Statistical analysis has been performed, and the result is shown in Table 2.
Reviewer 3 Report
Comments and Suggestions for Authors
The authors mainly addressed and corrected all concerns raised by the reviewer, therefore I suggest acceptance of this manuscript in its present form.
Author Response
Response to the Reviewer 3
Dear Reviewer,
Title: Phytochemical, technological and pharmacological study on the galenic dry extracts prepared from German chamomile (Matricaria chamomilla L.) flowers
Comment 1: The authors mainly addressed and corrected all concerns raised by the reviewer, therefore I suggest acceptance of this manuscript in its present form.
Response 1: Thank you very much for this positive statement.